

# Hotspots of sensitivity to GCM biases in global modelling of mean and extreme runoff.

Lamprini.V. Papadimitriou[1], Aristeidis G. Koutroulis[1], Manolis.G. Grillakis[1], Ioannis.K. Tsanis[1,2]

[1]Technical University of Crete, School of Environmental Engineering, Chania, Greece

[2]McMaster University, Department of Civil Engineering, Hamilton, ON, Canada

*Correspondence to*: I. K. Tsanis (tsanis@hydromech.gr)

**Abstract.** Climate model outputs feature systematic errors and biases that render them unsuitable for direct use by the impact models, especially when hydrological parameters are studied. To deal with this issue many

bias correction techniques have been developed to adjust the modelled variables against observations. For the most common applications, adjustment concerns only precipitation and temperature whilst for others more driving parameters (including radiation, wind speed, humidity, air pressure) are bias adjusted. Bias adjusting only a part of the variables required as biophysical model input could affect the physical consistency among input variables and is poorly studied. In this work we quantify the individual effect of bias correction

of each climate variable on global scale hydrological simulations of the recent past. To this end, a partial correction bias assessment experiment is conducted. Six climate parameters (precipitation, temperature, radiation, humidity, surface pressure and wind speed) from a set of three Global Climate Models are tested. The examined hydrological indicators are mean and extreme (low and high) runoff production. A methodology for the classification of the bias correction effects is developed and applied. Global hotspots of

hydrological sensitivity to GCM biases at the global scale are derived, for both mean and extreme runoff. Our results show that runoff is mostly affected by the biases in precipitation, temperature, specific humidity and radiation (in this order) and suggest that bias correction should be applied in priority to these parameters. Surface pressure and wind speed had a minor effect on runoff simulations for the majority of the land surface. Low runoff has an increased sensitivity to the GCM biases compared to mean and high runoff, underlying

the importance of bias correction for the study of low flow conditions and relevant hydrological extremes, such as droughts.

## 1  Introduction

In recent years, there is growing evidence of the changes in climate caused by the enhancement of global warming due to increased concentrations of anthropogenic green-house gas emissions (IPCC 2013; King et al. 2015). Under the pressing circumstances of a warming world, scientific research has focused on estimating the range of changes in the future climate and the effectiveness of different adaptation strategies. The main





tool for the investigation of future climate trends is the utilization of Global Climate Models (GCMs). GCMs are based on physical principles that describe the components of the climate system, such as cloud formation and water flux exchanges. Moreover, they are thoroughly evaluated against observed climate and past climate changes. Thus, there is considerable confidence on the ability of the models to simulate the changes in the

future climate (Randall et al., 2007). Although each generation of GCMs shows improvements compared to its predecessor, climate model outputs still contain substantial errors. These inherent biases can stem from the coarse resolution employed by the GCMs due to computing power limitations (Randall et al., 2007), from errors in the representation of physical atmospheric processes (Maraun, 2012) and from uncertainties regarding the boundary and initial model conditions (Bromwich et al., 2013). Biases are particularly present

in precipitation, the main driving variable of hydrological processes. GCM biases in precipitation and temperature are larger for the upper and lower (for temperature) tails of their distributions (Koutroulis et al., 2016). The systematic errors in hydrologically relevant GCM output variables constitute them unsuitable for use as forcing in impact models for hydrological and agricultural studies (Ehret et al., 2012), especially when extreme events are under assessment.

To overcome this limitation of the GCMs, various bias correction techniques have been developed. Bias correction can be described as the procedure of post-processing climate model data to statistically match observations. Typically bias correction methods are established based on a historic time period for which observations are available. The adjustments on the climate output prescribed by the bias correction method

are then applied to both historic and projected data. Therefore bias is considered to be stationary in time, one of the major assumptions of most bias correction methods (Hempel et al., 2013). Other drawbacks of bias correction are its dependency on the quality of observational data and the possible perturbation of feedbacks between the climate variables (Ehret et al., 2012; Hempel et al., 2013).

Bias correction procedures have mainly focused on adjusting the biases of precipitation and/or temperature (Christensen et al., 2008; Li et al., 2010; Piani et al., 2010), partly because many hydrological impact models only require these two meteorological variables as input. However, many state-of-the-art biophysical impact models –many participating in the InterSectoral Impact Model Intercomparison Project (ISI-MIP, Warszawski et al. 2014)- additionally need data on radiation, humidity, surface pressure and wind speed in

order to run, as they calculate both the water and energy budgets. The biases in these parameters can hinder the quality of the representation of hydrological fluxes such as runoff, evapotranspiration, snow accumulation and snowmelt by the impact models (Hagemann et al. 2011; Haddeland et al. 2012). Moreover, the use of uncorrected input variables together with bias corrected fields of precipitation and temperature may cause





inconsistencies in the energy balance calculations and introduce biases in the hydrological simulations (Hagemann et al., 2011; Rojas et al., 2011). Haddeland et al. (2012) investigated how the use of bias corrected radiation, humidity and wind speed in addition to bias corrected precipitation and temperature affects hydrological simulations. Their results showed that bias correction of the three additional climate parameters, brings simulations of the baseline period closer to observations.

Here, we investigate the effect of bias correcting climate parameters on the historical runoff output of a large scale hydrological model, with a focus on extreme events. Our study builds upon the study of Haddeland et al. (2012), who investigated the compound effect of bias correcting variables other than precipitation and temperature, by examining the effect of each climate variable individually. To the authors' knowledge, this is the first time that an assessment of each climate parameter's effect on hydrological output is performed. This study aims to bridge this knowledge gap in the literature by addressing the following research questions:

- To which extent does bias correction of climate model outputs contribute to a more consistent representation of past hydrologic indicators?
- Which forcing parameters affect the most and the least the hydrological output and how does this vary for different regions?
- Are there parameters that could be neglected from the bias correction procedure without significantly affecting the hydrological simulations?
- How does bias correction affect the simulation of hydrological extremes?

The remaining of this paper is organized as described below. Section 2 includes information on the hydrological model and the bias correction method used in this study along with the description of the performed experiment. In Section 3, the datasets used to force the hydrological model are briefly described. In Section 4 the results are presented and discussed. In the final Section 5, the study is summarized and conclusions are drawn.

## 2 Methods

### 2.1 Description of the impact model

Hydrological simulations were performed with the Joint UK Land Environment Simulator (JULES) model (Best et al., 2011). Examples of recent model applications can be found in Papadimitriou et al. (2016) and Grillakis et al. (2016). Here only a brief overview of the model is given. For a detailed description of JULES the reader can refer to the model description papers of Best et al. (2011) and Clark et al. (2011).





JULES is a physically based model that calculates water, energy and carbon exchanges between the land surface and the atmosphere. The science modules that comprise the model are: surface energy fluxes, snow cover and surface hydrology, soil moisture and temperature, soil carbon, vegetation dynamics and plant

physiology. These are connected through the physical processes that govern their interactions and feedbacks. The model requires seven climate variables as forcing, namely: precipitation, temperature, longwave and shortwave radiation, specific humidity, surface pressure and wind speed. Land cover heterogeneity is expressed by considering nine surface types, five of which represent vegetated areas while the rest correspond to non-vegetated areas. Each land grid-box is subdivided into a number of tiles that correspond to the different

surface types (Best et al., 2011). The energy budget is solved for each tile separately and the grid-box value is found by weighting the values of the tiles. Soil is represented by four soil layers of variable thickness (Best et al., 2011; Clark et al., 2011). This study analyzes the runoff production (RF) output of JULES. Runoff production has two components. The first one is surface runoff, produced by the infiltration excess mechanism. The second one is subsurface runoff (or drainage from the bottom of the soil column), which is

calculated as a Darcian flux under the assumption of zero gradient of matric potential. Calculation of potential evaporation follows the Penman–Monteith approach (Penman, 1948). Water held at the plant canopy evaporates at the potential rate while restrictions of canopy resistance and soil moisture are applied for the simulation of evaporation from soil and plant transpiration from potential evaporation (Best et al., 2011). Runoff is then converted to discharge at the basin outlet ($Q_{sim}$) through a delay algorithm (Papadimitriou et

al., 2016), to allow for comparison with discharge measurements ($Q_{obs}$).

### 2.2 Bias correction method

The bias correction methodology presented in Grillakis et al. (2013), namely MSBC, was used to adjust the biases of precipitation. The methodology follows the principles of quantile mapping correction techniques

and was originally designed and tested for GCM precipitation adjustment. The novelty of the method is the split of the data CDF space into discrete segments and then the individual quantile mapping correction in each segment, achieving better fit of the parametric equations on the data and thus better correction, especially on the CDF edges. The optimal number of the segments is estimated by Schwarz Bayesian information criterion SBIC to balance between complexity and performance. A modification of the

methodology was used for bias adjustment of the rest of the parameters that were used. Here the methodology is modified to use linear functions instead of the gamma that were used in the original methodology. Moreover, the edge segments are explicitly corrected using only the difference between the historical period model data and the observations. This choice costs to the methodology the remain of some bias in the





corrected data. It provides however rigidity, avoiding unrealistic temperature values at the edges of the model CDF.

The bias adjustment methodology modification has been already used in the Bias Correction Intercomparison

Project (BCIP) (Nikulin et al., 2015), while adjusted data of the methodology have also been used in a number of climate change impact studies (Grillakis et al., 2016; Papadimitriou et al., 2016). As MSBC methodology belongs to the parametric quantile mapping techniques, it shares their advantages and drawbacks. A comprehensive analysis of advantages and disadvantages of the methods that follow the quantile mapping comparing to others can be found in Maraun et al. (2010) and Themeßl et al. (2012).

## 2.3 Experimental design

To serve the purpose of examining the effects of each forcing parameter's bias on the runoff output, an experiment comprised of nine sets of JULES' runs was designed. A graphical description of the performed experiment is shown in Figure 1. The time span of this analysis is the baseline period 1981-2010. This is also

the time span of the period used for bias correction of the GCM output. Climate data from 3 GCMs and the WFDEI dataset are used as JULES' forcing. The sets of runs forced with GCM data, include three model runs –one per GCM. Then the analysis progresses using the ensemble mean.

The first part of the experiment is bias assessment and includes three sets of JULES' runs:

i)      forced with WFDEI (WFDEI)

ii)     forced with uncorrected climate data (RAW)

iii)    forced with bias corrected climate data (BC).

These are used to assess initial and remaining biases in the forcing data themselves and in the resulting

hydrological simulations. The term "initial bias" is used to describe the difference between uncorrected meteorological variables and the respective WFDEI variables. "Remaining bias" is the difference between bias corrected meteorological variables and the respective WFDEI variables. When referring to runoff, "initial" and "remaining" biases are the difference between runoff simulations forced with uncorrected and corrected forcing respectively from simulations forced with the WFDEI dataset.

The second part of the experiment is the partial correction bias assessment. For this part, six more sets of JULES' runs were performed. In each of these runs, one of the six forcing variables (precipitation, temperature, radiation, humidity, surface pressure and wind speed) is used in its raw form while the rest of





the input forcing is bias corrected. Runoff from the runs with partially corrected input are compared to runoff from the run for which all input variables are bias corrected. This comparison allows us to assess the "loss" in the quality of simulations when a parameter is neglected from the bias correction procedure. It must be noted however that the "loss in quality" concept bears the assumption that the simulated runoff from a fully

corrected set of forcing variables is of better quality and closer to observations compared to a partially corrected set.

A clarification is needed regarding radiation, as it has been previously mentioned that JULES requires two separate components of radiation as input -longwave and shortwave. For the purposes of this experiment we

chose to examine the effect of the radiation flux as a compound rather than the effect of each radiation component. Thus, in the run for which radiation is examined, both shortwave and longwave radiation were considered unadjusted.

The hydrological indicators studied are: mean, low and high runoff production. Low runoff is approached

using the 5th percentile runoff (or Q5) and high runoff using the 95th percentile runoff (Q95). When referring to both low and high runoff, the term "extreme" runoff is used. The hydrologic indicators are derived for the 1981-2010 period.

## 2.4 Categorization of bias correction effects

A framework is developed to classify the effect of input forcing biases on output runoff in three qualitative tiers: weak, moderate and strong effect (shown in Figure 2). Here biases in the input forcing are the difference between RAW and BC variables ($\Delta Var=RAW\_Variable–BC\_Variable$). The effect of a forcing variable's bias on runoff is found from the difference between runoff forced with all bias corrected forcing except for the variable of interest and runoff from all bias corrected data ($\Delta Q=RF$ forced with NoBCVar-RF forced with

BC). $\Delta Var$ and $\Delta Q$ are expressed as percentages. The only exception is for temperature, where $\Delta Var$ refers to the absolute difference between raw and bias corrected temperature in degrees. Handling change as a percentage is a way of normalizing the effect of each variable into the same unit, thus allowing us to compare the effect of different variables under a common scale.

Bias's effect is categorized by creating a scatterplot of $\Delta Q$ versus $\Delta Var$. According to the developed framework shown in Figure 2 the scatterplot area is divided in three areas that correspond to weak, moderate or strong effect of the variable's bias on runoff. "Strong" effect means that a relatively small bias in the input variable causes a large change in runoff. Inversely, the effect is "weak" when large biases in the input variable result in small $\Delta Q$ and also when both $\Delta Var$ and $\Delta Q$ are small. The boundaries set to distinguish between the



levels of the effect are, of course, arbitrary and would possibly need modifications if applied in different kinds of studies.

On the implementation of the described framework, each grid-box receives a label describing the effect of
the bias, according to where its ΔVar and ΔQ values fall in the scatterplot. Then the labeled grid-boxes are assigned to their geographical location. The result is maps of bias effect tiers for each examined forcing variable.

It must be noted that although the scatterplot in Figure 2 covers the -100 to 100% range for both the x and y
axis, practically the values can be higher than the shown limit of 100%. The implemented framework expands to higher values. For a clearer and more understandable presentation of the established method, the scatterplot shown here is limited to the 100% value.

## 2.5 Regional scale bias assessment

Input forcing biases and their effects are also investigated regionally. Focus is given at 24 regions, which were selected from the 26 regions introduced by Giorgi & Bi (2005) (in our study Alaska and Greenland are excluded from the analysis). The selected regions and their abbreviations are shown in  Table *1*.

## 2.6 Hydrological evaluation

For the evaluation of JULES' hydrological performance, three metrics are used: Nash-Sutcliffe efficiency (NSE), Percent bias (PBIAS) and the coefficient of determination ($R^2$). The formulas for the calculation of NSE and PBIAS are given below:

$$NSE = 1 - \left[ \frac{\Sigma(Q_{sim} - Q_{obs})^2}{\Sigma(Q_{obs} - Q_{mean})^2} \right] \qquad (1)$$

$$PBIAS = \left[ \frac{\Sigma(Q_{sim} - Q_{obs}) * 100}{\Sigma Q_{obs}} \right] \qquad (2)$$

In the above equations, $Q_{sim}$ corresponds to the simulated values, $Q_{obs}$ to observations and $Q_{mean}$ to the mean of observed data.

The evaluation metrics presented in this paper are calculated based on data of seasonal monthly (annual cycle) time resolution. Discharge measurements obtained from the Global Runoff Data Centre (GRDC) database were used as observations.



### 3 Climate data

The climate dataset used in this study for bias correction of the GCM output and as a baseline for comparison of the results is called the WATCH Forcing Data methodology applied to ERA-Interim data (WFDEI; Weedon et al. 2014). WFDEI data span from 1979 to 2012, but here only the time period from 1981 to 2010

5 was used. The WFDEI dataset is based on its predecessor WFD (WATCH Forcing Data; Weedon et al. 2010), which was derived from the ERA-40 reanalysis product (Uppala et al., 2005) after interpolation to half-degree resolution, elevation adjustments and monthly-scale corrections made against gridded station observations of the Climate Research Unit. For detailed information on the derivation of the WFDEI dataset the reader is referred to Weedon et al. (2014).

Data from three GCMs participating in the fifth phase of the Coupled Model Intercomparison Project (CMIP5; Taylor et al. 2012) were used as forcing. Information on the ensemble members can be found in Table 2. Climate model outputs were interpolated to the $0.5^{\circ}$ spatial resolution of the WFDEI dataset, using the nearest-neighbor method.

### 4 Results and Discussion

### 4.1 Biases in forcing variables at the global scale

Global maps of the initial and remaining biases of the forcing variables are shown in Figure 3. In general terms the remaining biases are smaller than the initial ones by one to two orders of magnitude.

For precipitation (Figure 3a), wet biases are prevalent at the continents of Europe, Asia, Africa and Oceania as well as at the west part of the American continent. The largest wet biases are observed for regions with high mountain ranges (the Andes in South America, the Alaska Range and the Rocky Mountains in North America and the Himalayas in Asia) and for the tropical African and Indonesian regions. The most dominant

25 dry biases are found in the Amazon region. The biases that remain in precipitation after bias adjustment are small (up to 0.01 mm/day for the most part of the land surface), mostly negative and located in the tropics.

The raw GCM ensemble exhibits a cold bias in temperature over the African continent, northern Europe, central and east Asia and western-north America (Figure 3b). Warm biases are found in the Alaskan,

30 Greenland, north and central Asia regions as well as in the Mediterranean and the Andes. After bias adjustment, the temperature bias is less than 0.1 degrees K for almost all the land surface.





Generally, the GCM biases in precipitation and temperature are more pronounced over high mountainous regions and the tropics. Recent studies argue towards a dependency between biases and altitude. According to the study of Haslinger et al. (2013), both temperature and precipitation biases of a GCM tested over the Alpine Region, show increasing trends with height. Palazzi et al. (2016) found increasing warming rates at

higher altitudes of the Tibetan Plateau-Himalayan region, examining the temperature of an ensemble of twenty-seven CMIP5 models. Regarding the tropics, various studies show increased GCMs' biases in these regions compared to model performance in other climate zones (Koutroulis et al., 2016; Randall et al., 2007; Solman et al., 2013).

The bias of longwave radiation is negative over South America, Africa, south and south-east Asia, north-east Europe and north Asia (Figure 3c). In contrast, the biases in shortwave radiation are positive for the most part of the land surface (Figure 3d). The remaining bias of both long- and short- wave radiation ranges from -0.1 to 0.1 W/m$^2$. Specific humidity exhibits positive biases in north America, Europe, central and north Asia, Indonesia, Oceania and south Africa (Figure 3e). The biases that remain after the correction procedure are

both positive and negative, ranging between $-10^{-5}$ to $10^{-5}$ kg/kg. Surface pressure is the variable with the most variable pattern in the initial bias (Figure 3f). It is therefore difficult to identify regions of over- and under-estimation of the variable by the raw GCM ensemble. Nonetheless, the areas where high mountain ranges are located (Rocky Mountains, Andes, Himalayas) distinguish as regions with considerable positive biases in surface pressure. The remaining bias in surface pressure is less than 0.1 HPa (in absolute terms) for most of

the land surface. Wind's initial biases are mainly negative (Figure 3g). Regions where positive biases are found include west-north America, the Mediterranean along with central and northeastern Asia. The remaining biases of the wind variable range between -0.01 and 0.01 m/s.

### 4.2 Regional and seasonal biases in forcing variables

The next step of this analysis includes the spatial integration of the gridded information on forcing biases. The results are shown in Figure 4, which illustrates the biases of the raw GCM ensemble in comparison to the WFDEI dataset, averaged over 24 regions of the globe. To account for possible seasonality variations in the biases, the differences are calculated for the annual mean (ANN) and for the December-January-February (DJF) and June-July-August (JJA) means.

The wettest precipitation biases are encountered in the equatorial and Southern Africa (EQF, SQF and SAF) and concern the DJF precipitation (Figure 4). The driest biases are found for the CAM, AMZ and SAS regions, for the JJA precipitation. Precipitation biases are less pronounced in Europe (NEU, MED, NEE) and in central and north Asian regions (CAS, NAS). Temperature displays cold biases in most regions. A notable





exception is the warm bias in DJF temperature in the NAS region, which is the most pronounced temperature bias found. Generally the DJF temperature biases are the largest, followed by ANN while the JJA season has the smallest temperature biases.

The two radiation components, long-wave (Rlds) and short-wave (Rsds) radiation, show an inverse behavior in their biases (Figure 4). That is to say, in regions where Rlds has negative biases Rsds shows positive biases and vice versa. Negative biases are dominant for Rlds in contrast to the Rsds variable which mostly shows positive biases. Specific humidity has negative biases over the north part of the African continent (SAH, WAF, EAF, EQF), central and south America (CAM, AMZ, CSA) and south Asia (SAS). Positive humidity

biases are identified in the south part of Africa (SQF and SAF) and north America (WNA, CNA and ENA).

Surface pressure shows almost exclusively positive biases (Figure 4). The regions that distinguish for the largest biases are MED, SEA, SAH, SAF, CAM, CSA and SSA. The most dominant negative wind speed bias is found in NAU. Most of the African continent (SAH, WAF, EAF, EQF, SQF) and of South America

(AMZ, CSA) also have negative biases in wind. The largest positive biases are encountered in the southern part of South America (SSA) for the JJA season and for the DJF season in regions of North America (WNA, CAM), Europe (MED) and Asia (CAS, TIB, SEA).

**4.3 Model validation**

In order to assess model performance, simulated discharge forced with the WFDEI dataset is compared to observations. Additionally, discharge simulations of the raw GCM dataset are included in the comparison to quantify the biases present in discharge due to forcing data bias propagation through the JULES model.

Figure 5 shows the seasonality of discharge for nine study basins, as simulated by the model when forced

with WFDEI and raw GCM data in comparison to measured discharge. The evaluation metrics of the two sets of simulations are presented in Figure 6. Discharge seasonality is calculated for the 1981-2010 time period. For seven out of the nine basins, seasonality is well captured by the WFDEI simulation (Figure 5). In contrast, the raw GCM simulation exhibits significant positive and negative biases for these seven basins. For the two remaining basins however (Mississippi and Lena) seasonality is best captured by the raw GCM

simulation. The WFDEI run results in positive NSE values (0.24 to 0.94) for all the basins. On the contrary the raw GCM run results in negative NSE values for six out of the nine basins. The PBIAS index indicates that the raw GCM simulation has greater deviations from observations than the WFDEI run for most basins (exceptions are Mississippi, Lena, Ganges and Danube). Finally, the $R^2$ metric shows that the linear





correlation between simulations and observations is stronger for the WFDEI run for seven out of the nine basins (exceptions are Mississippi and Elbe). For both simulations the lowest $R^2$ value is reported for the Congo basin (0.45 and 0.2 for the WFDEI and raw GCM runs respectively). The best correlations per simulation are found for Ganges for the WFDEI run (0.99) and for Amazon for the raw GCM run (0.94).

Here it has to be noted that the persistent departure from the mean climatology of discharge includes four types of errors. First, it includes the error in the bias correction or remaining bias and secondly the insufficient description of the runoff processes by the land surface model. Additionally, the two other types of error are the error in the observational datasets with regards to depicting the real climatic drivers and the error in runoff measurements.

### 4.4 Biases in output runoff at the global scale

Figure 7 shows the initial and remaining biases in output runoff, in comparison to the WFDEI run. Results are shown for ANN, DJF and JJA means. As with the biases in the input forcing variables, the remaining bias in runoff is one to two orders of magnitude smaller than the initial bias.

Regarding the raw GCM run, the largest runoff underestimation biases (<-5 mm/day) are encountered in central-north America, the central-east part of South America and East Asia. The most pronounced runoff overestimation biases are found in the west part of North and South America, in equatorial, south Africa, northern Europe, the Tibetan region and Indonesia. The differences between the seasonal means and the

annual mean are in general subtle. Yet, the increases in DJF runoff overestimation in south equatorial Africa and in JJA runoff in the Tibetan plateau are worth noting.

The remaining biases in runoff range from -0.1 to 0.1 mm/day for most of the land surface. Apart from having been reduced significantly, the remaining biases' sign is inversed compared to the initial biases. Areas where

the (negative) remaining bias in ANN runoff is more pronounced are the west Amazonian region and Indonesia. Concerning seasonal runoff, there are also remaining DJF biases in Australia and JJA biases in equatorial Africa and eastern Asia.

The use of bias corrected data led to an improved representation of mean runoff by the model. Accordingly,

the studies of Teutschbein & Seibert (2012) and Rojas et al. (2011) found that hydrological simulations are substantially improved with the use of bias corrected forcing.



### 4.5 Effect of each forcing parameter's bias on runoff at the global scale

To assess the effect that each forcing parameter's bias has on runoff, runoff from the simulations where one forcing parameter is neglected from bias correction is compared to runoff from the run with all the forcing variables bias adjusted. Results are shown in Figure 8, which illustrates the differences between runoff from

each experiment and runoff from all bias corrected input forcing, for ANN, DJF and JJA averages.

Figure 8 indicates that precipitation is the parameter that mostly affects runoff, in comparison to the other forcing variables. Precipitation bias causes both wet and dry biases in different regions of the land surface, with a pattern and values that closely resemble the effect of the initial GCMs' biases on runoff (Figure 7). A

similar pattern between precipitation and runoff biases was also observed by Teng et al. (2015), who noted that precipitation errors are magnified in modelled runoff. Temperature biases result in runoff overestimation over west- and east-North America, the Amazon region, equatorial Africa, northern Europe and parts of Asia (north, south and south-east Asia). Smaller in extent are the regions where temperature biases produce a negative bias in runoff (Greenland, central-south America, parts of the Mediterranean and of central Asia).

Excepting the radiation components from the bias correction procedure results in runoff underestimations by the model over the regions of west-north America, Alaska and Greenland, central and most of South America, west equatorial Africa, most of Europe and north Asia along with south and south-east Asia. The bias of the specific humidity variable causes runoff overestimation in the higher latitudes (northern part of north

America, Europe, north Asia) and in the Indonesia region. However, in central and South America, central equatorial Africa and in the mountainous regions of south- and south-east Asia, specific humidity bias results in runoff overestimation. Other studies also advocate towards the considerable effect that biases in radiation (Mizukami et al., 2014) and humidity (Masaki et al., 2015) can have on hydrological fluxes.

Surface pressure and wind are the variables whose bias affects the least the hydrological output. For surface pressure, it can be observed that the variable's bias causes a negative bias in runoff simulations in the high mountain ranges' regions of South America and Asia (Andes and Himalayas respectively). Wind biases result in runoff overestimations over the west parts of North America, over South America and equatorial Africa and also over north Europe, northwest and southeast Asia.

### 4.6 Regionalized effect of each forcing parameter's bias on runoff

In the following step of this analysis, the relationship between the biases in input forcing and output runoff is investigated per land region. For all the land grid-boxes that are included into the 24 regions' boundaries,



the input variable bias and the output runoff bias are plotted against each other in a scatterplot. The results for 10 selected regions, along with the location and the extent of the selected regions, are shown in Figure 9. The presented regions are selected as representative of different parts of the land surface, as the number of the regions shown in the manuscript had to be reduced for clarity of the results. Scatterplots of the 24

examined regions can be found in the Supplement of this paper. The median values of the percent bias in input forcing and the respective percent change in runoff for the 24 examined regions examined are shown in Table 3.

The correlation between biases in precipitation and changes in runoff resembles a directly proportional linear

relationship. This behavior is more pronounced in some regions (particularly ENA and NEU) and more obscured in others where the data cloud appears more scattered (MED, WNA) (Figure 9). According to the median values of the changes in Table 3, some regions are dominated by negative changes in precipitation (MED, SAS, AMZ, CSA) and others by positive biases (NEU, WNA, ENA, CAM, WAF, SAU). The regions with the largest precipitation biases and consequent runoff changes are WNA ($\Delta var$=65.92%, $\Delta q$=112.66%),

CSA ($\Delta var$=-32.8%, $\Delta q$=-63.21%), WAF ($\Delta var$=26.74%, $\Delta q$=58.24%) and AMZ ($\Delta var$=-26.58%, $\Delta q$=-40.52%). Temperature biases have an inversely proportional relationship with changes in runoff. Nonetheless this relationship is quite variant for different regions. For example, in NEU, small temperature biases (median $\Delta var$=-0.46 K) result in large changes in runoff (median $\Delta q$=22.68%). In contrast, in CAM, almost double the temperature bias (median $\Delta var$=-0.98 K) causes only 3.65% median change in runoff.

Specific humidity biases also affect in various ways the runoff output in the different geographic regions examined. In NEU, MED, WNA and ENA, a small range of positive biases results in a high range of positive runoff changes. A different behavior is observed for CAM, SAS, AMZ and CSA where the data cloud is more scattered on the x axis (meaning larger biases in specific humidity) and less scattered on the y axis (i.e.

changes in runoff are smaller). Radiation biases cover a small range but the effect on runoff is much larger and can reach up to (or even exceed) 100% for some regions (WNA, SAS, WAF, AMZ). Exceptions are ENA, CAM and SAU, as for these regions both the forcing biases and runoff changes are very small. The largest humidity bias and the largest consequent runoff change are reported for AMZ ($\Delta var$=4.06%, $\Delta q$=-9.34%).

Surface pressure has smaller biases compared to the other forcing variables (minimum median is -0.05% for NEU) and its effect on runoff also appears reduced. Finally, wind has a wide range of both positive and





negative biases (minimum median is -15.13% for NEU and maximum is 25.27% for CAM) which, however, do not seem to affect runoff in a consistent manner.

### 4.7 Hotspots of mean and extreme runoff sensitivity to forcing biases

Figure 10 visualizes the categorization of each forcing variable's bias effect on runoff, at the global scale. Moreover, the land area fraction corresponding to each effect category is tabulated in Table 4.

Precipitation is the variable with the largest faction of strongly affected area (15.39%), followed by specific humidity (13.82%) and temperature (11.69%). Regions where a strong effect of precipitation's bias is

encountered are the western parts of North and South America, west Africa and parts of central Europe and Asia. For specific humidity, the strongly affected areas show a significant spatial coherence and are clustered in the higher latitudes of the globe (North America, central east Europe, central and north Asia). The areas strongly affected by temperature biases are scattered around North and South America, north east Europe, central, south and north Asia. Precipitation is also the variable with the largest land fraction corresponding

to a moderate effect of its bias on runoff (66.71%), followed by temperature (40.78%) and radiation (32.55%). Surface pressure and wind biases have a weak effect on runoff for the vast majority of land area (92.13% and 90.19% respectively).

Next, the analysis focuses on the changes in the effect category pattern when the effects on low and high

runoff (rather than mean runoff) are examined. In Figure 11, the areas that are additionally strongly affected by forcing biases, when the analysis is switching from the effect on the mean to the effect on extreme low and high runoff, are shown. Table 4 gives information on the land fraction that is additionally affected when examining effects on low and high runoff.

A first observation is that forcing biases affect strongly low runoff in considerably more area compared to mean runoff. When examining high runoff, the extent of the affected area is only slightly increased compared to mean runoff. For low runoff, the affected area shows the largest increase in response to precipitation biases (+22.97%), followed by temperature (+18.74%) and specific humidity (+14.24%). Considering the very small land fraction that showed effects on mean runoff, substantial increases in area where low runoff is

strongly affected are denoted for surface pressure and wind (reaching 6.11% and 7.19% respectively). The area where high runoff is strongly affected by forcing biases shows the greatest increase for temperature (+3.40%) followed by precipitation (+3.15%). Surface pressure and wind give the smallest increases compared to the other input variables but still the total fraction of strongly affected area for high runoff is





more than double the number for mean runoff (1.10% and 1.11% respectively). High sensitivity of low flows to bias correction along with a small effect of bias correction on high flows was also reported in the study of Muerth et al. (2013). According to Muerth et al. (2013), high flows show insensitivity to bias correction because their simulation is mainly governed by other parameters, namely the structure of the hydrological model and the frequency of extreme precipitation events.

### 4.8 Study caveats

An issue that must be considered for the interpretation of the results of this study is that they have been based on a single impact model. As the uncertainty stemming from the selection of the impact model is large (Gudmundsson et al. 2012; Hagemann et al. 2013), it is preferable to use multiple models in order to capture a wide range of possible results. The effect of the meteorological forcing on a hydrological output is heavily model dependent, as different models employ different concepts and/or equations for the representation of key hydrological processes. This concern has been also discussed by other single model studies on meteorological variables' effects on hydrological outputs (Mizukami et al. 2014; Masaki et al. 2015). Nonetheless, the results of single model studies are useful in giving indicative answers on the issues they examine and set a basis for the methodology that would be needed for respective multi-model applications.

### 5 Summary and conclusions

The present study examined the effect of bias correcting GCM output variables on mean and extreme (low and high) runoff simulations of the recent past. Bias's effects were studied for each forcing variable separately, for a total of six meteorological parameters (precipitation, temperature, radiation, specific humidity, surface pressure and wind speed). A framework for the comparison and categorization of the effects of biases of the different variables was developed. The method was implemented for mean, low and high runoff, leading to maps of sensitivity to biases and identification of sensitivity hotspots. The conclusions derived from this work are presented below.

- Bias correction of climate model outputs results to substantially improved representation of past hydrologic indicators. For this reason, our study adds to the numerous studies that advocate on the use of some kind of bias correction of GCM data prior to their use for hydrological applications and climate impact assessments.
- Precipitation, as expected, is the parameter that mostly affects runoff. Temperature and specific humidity follow, but their effect mostly applies to the northern hemisphere. Radiation has a moderate effect on



runoff in many areas of the globe while surface pressure and wind speed have only a weak effect on runoff for the vast majority of the land surface.

- This study indicates that the widely used concept of bias correcting precipitation and temperature should
be extended to include more input variables. Based on our findings, the suggested priority parameters for bias correction in hydrological applications are precipitation, temperature, specific humidity and radiation (in that order).

- Bias correction does not affect high runoff considerably more than mean runoff. In contrast, low runoff
exhibits an increased sensitivity to the GCM biases. Thus bias correction is even more important when studying events relevant to low flow conditions, such as droughts.

## Acknowledgements

"We acknowledge the World Climate Research Programme's Working Group on Coupled Modelling, which is responsible for CMIP, and we thank the climate modeling groups (listed in Table 2 of this paper) for producing and making available their model output. For CMIP the U.S. Department of Energy's Program for Climate Model Diagnosis and Intercomparison provides coordinating support and led development of software infrastructure in partnership with the Global Organization for Earth System Science Portals."

The research leading to these results has received funding from HELIX project of the European Union's Seventh Framework Programme for research, technological development and demonstration under grant agreement no 603864.

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



**Table 1. 24 regions of the globe, selected from Giorgi & Bi (2005).**

| Full region name | Abbreviation |
|---|---|
| North Europe | NEU |
| Mediterranean Basin | MED |
| Northeast Europe | NEE |
| North Asia | NAS |
| Central Asia | CAS |
| Tibet | TIB |
| Eastern Asia | EAS |
| Southeast Asia | SEA |
| Northern Australia | NAU |
| Southern Australia | SAU |
| Sahara | SAH |
| Western Africa | WAF |
| Eastern Africa | EAF |
| East Equatorial Africa | EQF |
| South Equatorial Africa | SQF |
| Southern Africa | SAF |
| Western North America | WNA |
| Central North America | CNA |
| Eastern North America | ENA |
| Central America | CAM |
| Amazon | AMZ |
| Central South America | CSA |
| Southern South America | SSA |
| South Asia | SAS |





**Table 2. Information on the GCMs used for this study.**

| Modelling group | Institute ID | Model name | Horizontal Atm. Resolution (lon x lat) | Key reference |
|---|---|---|---|---|
| Institut Pierre-Simon Laplace | IPSL | IPSL-CM5A-LR | 3.75 x 1.88 | Dufresne et al. (2013) |
| Japan Agency for Marine-Earth Science and Technology, Atmosphere and Ocean Research Institute (The University of Tokyo), and National Institute for Environmental Studies | MIROC | MIROC-ESM-CHEM | 2.81 x 2.81 | Watanabe et al. (2011) |
| US Dept. of Commerce/NOAA/Geophysical Fluid Dynamics Laboratory | GFDL-NOAA | GFDL-ESM2M | 2.50 x 2.00 | Dunne et al. (2012) |



**Table 3. Percent bias of input variable and resulting change in output runoff. The median value of all gridboxes of 24 land regions are shown.**

|  | % Difference | Pr [kg/m2/s] | Tas [K] | Huss [kg/kg] | Rlds/Rsds [W/m2] | Ps [Pa] | Wind [m/s] |
|---|---|---|---|---|---|---|---|
| GLOBAL | Variable | 14.46 | -0.57 | 0.91 | 1.73 | -0.02 | -5.86 |
|  | Runoff | 2.49 | 3.38 | 2.04 | -3.71 | -0.04 | 0.21 |
| NEU | Variable | 14.60 | -0.46 | 4.10 | 1.86 | -0.05 | -9.79 |
|  | Runoff | 27.97 | 22.68 | 25.49 | -5.25 | -0.02 | 3.62 |
| NEE | Variable | 4.89 | -1.44 | 3.32 | 2.44 | 0.10 | -11.77 |
|  | Runoff | 5.75 | 47.11 | 32.73 | -5.39 | 0.26 | 5.98 |
| NAS | Variable | 26.05 | 0.67 | 8.05 | 3.53 | -0.06 | -1.08 |
|  | Runoff | 59.36 | 11.80 | 63.98 | -10.08 | 0.02 | 4.06 |
| WNA | Variable | 65.92 | -1.75 | 13.55 | -1.23 | 0.14 | 10.23 |
|  | Runoff | 112.66 | 17.94 | 9.85 | -0.48 | 0.16 | -2.50 |
| CNA | Variable | -12.84 | 0.11 | 2.29 | 1.68 | -0.08 | -14.79 |
|  | Runoff | -50.86 | 1.53 | 6.57 | -2.06 | -0.05 | 1.96 |
| ENA | Variable | 4.08 | 0.49 | 13.40 | 2.71 | 0.10 | 5.47 |
|  | Runoff | -0.38 | -0.38 | 39.72 | -5.18 | 0.13 | 0.86 |
| MED | Variable | -14.39 | -0.15 | -1.34 | 0.55 | 0.41 | 14.94 |
|  | Runoff | -58.56 | 1.55 | 4.07 | -1.51 | 0.44 | -0.47 |
| CAS | Variable | 6.44 | -0.03 | -13.00 | 1.37 | -0.41 | 8.09 |
|  | Runoff | -9.94 | 1.31 | -0.19 | -0.44 | -0.36 | -1.29 |
| TIB | Variable | 128.47 | -2.94 | 7.69 | -1.14 | -0.12 | 12.59 |
|  | Runoff | 1017.17 | 5.38 | 0.81 | 0.97 | 0.02 | 0.06 |
| EAS | Variable | 19.25 | -0.94 | 2.92 | 2.51 | -0.20 | -3.55 |
|  | Runoff | 4.36 | 5.54 | 3.66 | -2.96 | -0.05 | 0.76 |
| CAM | Variable | 11.43 | -0.98 | -6.16 | -0.40 | 0.15 | 25.27 |
|  | Runoff | -7.73 | 3.65 | -2.55 | -0.10 | 0.14 | -0.52 |
| SAH | Variable | 54.11 | -2.73 | -8.96 | -0.47 | 0.22 | -13.59 |
|  | Runoff | -2.59 | -0.68 | -0.32 | 0.64 | 0.00 | 0.08 |
| SAS | Variable | -9.19 | -1.08 | -13.11 | 1.39 | -0.05 | -6.81 |
|  | Runoff | -26.35 | 5.20 | -2.53 | -4.07 | -0.09 | 0.51 |
| WAF | Variable | 26.74 | -1.51 | -5.79 | -0.88 | -0.10 | -15.13 |
|  | Runoff | 58.24 | 5.61 | -0.71 | -1.57 | -0.13 | 0.09 |
| EAF | Variable | 23.22 | -1.68 | -5.76 | -0.06 | -0.25 | -12.11 |
|  | Runoff | 42.13 | 7.24 | -3.74 | -1.51 | -0.28 | 0.09 |
| EQF | Variable | 5.64 | -1.55 | -2.15 | -0.25 | -0.20 | -10.09 |
|  | Runoff | -0.14 | 6.21 | -1.29 | 0.92 | 0.00 | 0.07 |
| SEA | Variable | 19.76 | -0.87 | 0.89 | 1.11 | 0.23 | 34.57 |
|  | Runoff | 43.92 | 5.97 | 1.66 | -3.20 | 0.32 | -1.04 |



| | | | | | | | |
|-----|----------|--------|--------|--------|--------|--------|--------|
| AMZ | Variable | -26.58 | -0.35 | -13.19 | 4.06 | -0.19 | -4.00 |
| | Runoff | -40.52 | 4.88 | -6.01 | -9.34 | -0.23 | 0.03 |
| SQF | Variable | 36.45 | -0.90 | 0.89 | 0.90 | -0.03 | -15.60 |
| | Runoff | -73.18 | -82.26 | -84.68 | -85.07 | -84.20 | -84.18 |
| NAU | Variable | 41.15 | -0.04 | 7.71 | 1.43 | 0.10 | -28.46 |
| | Runoff | -5.13 | 1.02 | 1.38 | -1.16 | 0.09 | -0.44 |
| CSA | Variable | -32.80 | 0.70 | -11.53 | 3.05 | -0.23 | -7.50 |
| | Runoff | -63.21 | -1.49 | -5.75 | -3.22 | -0.13 | 0.38 |
| SAF | Variable | 89.80 | -1.41 | 14.28 | -0.38 | 0.68 | -4.74 |
| | Runoff | 85.47 | 5.50 | 5.33 | 0.54 | 0.42 | -0.02 |
| SAU | Variable | 18.92 | -0.28 | 2.00 | 0.85 | -0.13 | -11.20 |
| | Runoff | -9.29 | 1.07 | 1.40 | -0.11 | 0.06 | -0.49 |
| SSA | Variable | 72.07 | -1.22 | 5.07 | -1.77 | 0.08 | 9.91 |
| | Runoff | 84.32 | 10.06 | 12.05 | -0.47 | 0.34 | -2.44 |



**Table 4. Percent of land area under each forcing variable's effect category (Weak, Moderate or Strong).**

|  | Pr | Tas | R | Huss | PS | Wind |
|---|---|---|---|---|---|---|
|  | Mean runoff | | | | | |
| Weak | 17.90% | 47.53% | 63.19% | 58.69% | 92.13% | 90.19% |
| Moderate | 66.71% | 40.78% | 32.55% | 27.49% | 7.41% | 9.48% |
| Strong | 15.39% | 11.69% | 4.26% | 13.82% | 0.46% | 0.33% |
|  | Change in percent area for Low runoff | | | | | |
| Strong | +22.79% | +18.74% | +9.88% | +14.24% | +5.65% | +6.86% |
|  | Change in percent area for High runoff | | | | | |
| Strong | +3.15% | +3.40% | +2.07% | +2.70% | +0.64% | +0.78% |





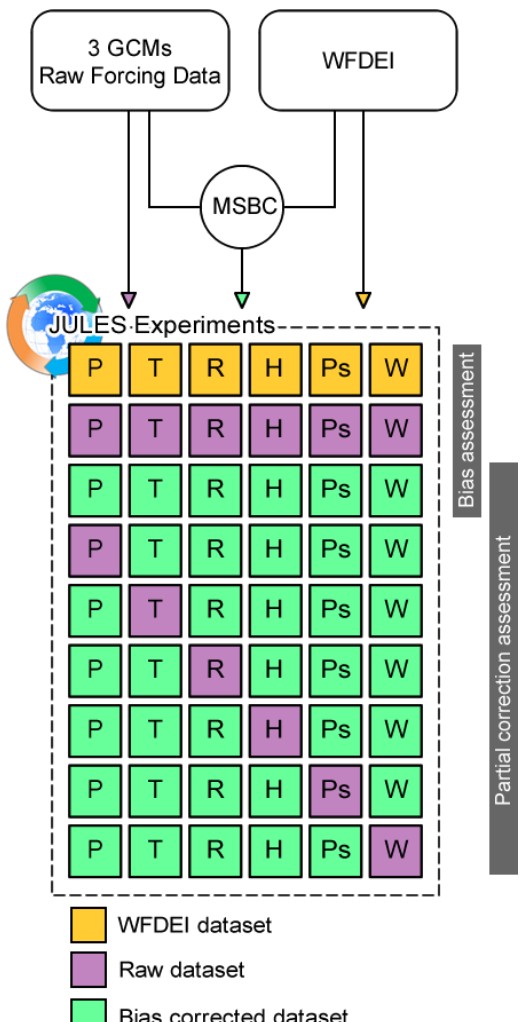

**Figure 1. Description of the performed experiment.**




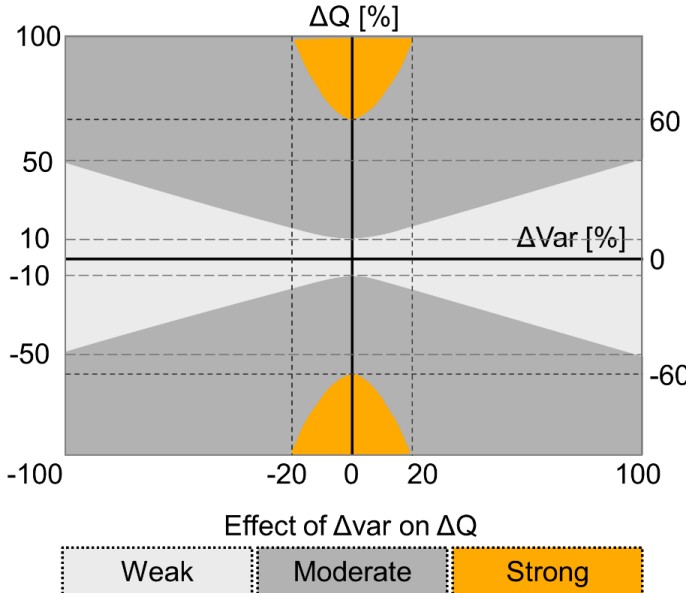

**Figure 2. Categorization of effect of change in forcing variable (Δvar) on change in runoff (ΔQ) as "Weak", "Moderate" or "Strong".**





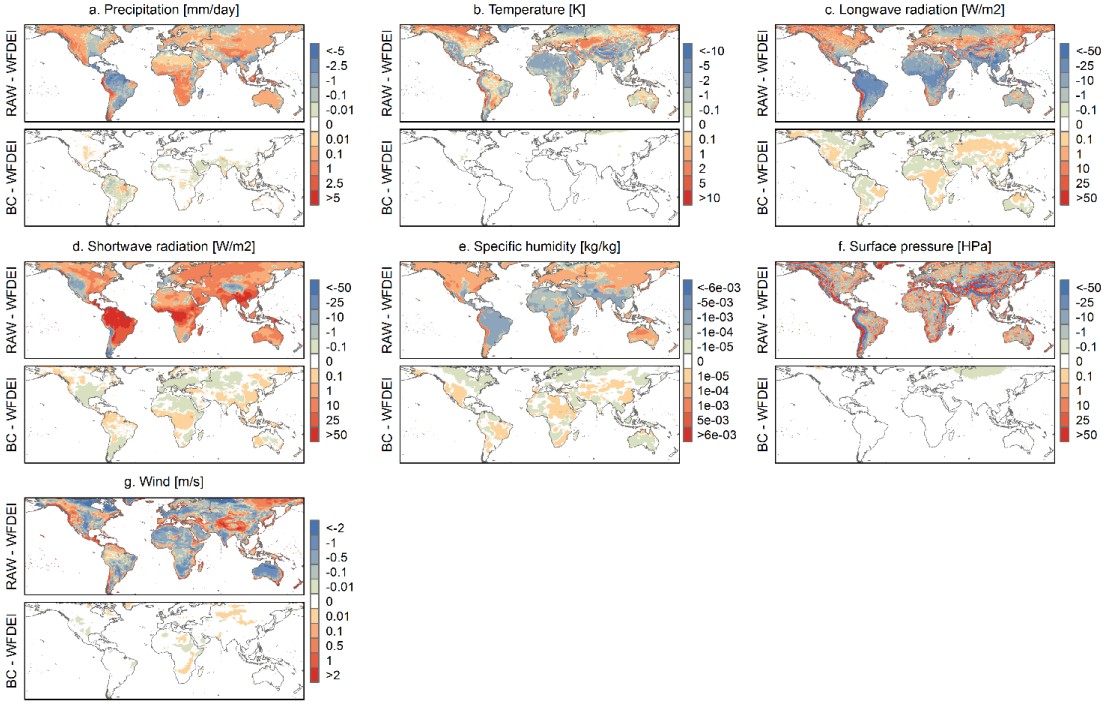

**Figure 3. 1981-2010 Annual Averages (ANN) of Forcing Data, differences of WFDEI from raw and bias adjusted input.**





**Figure 4. Differences between the forcing variables of the raw GCM ensemble and the WFDEI dataset. Annual (ANN), December-January-February (DJF) and June-July-August (JJA) averages for the period 1981-2010, spatially averaged for 24 Giorgi regions are shown.**



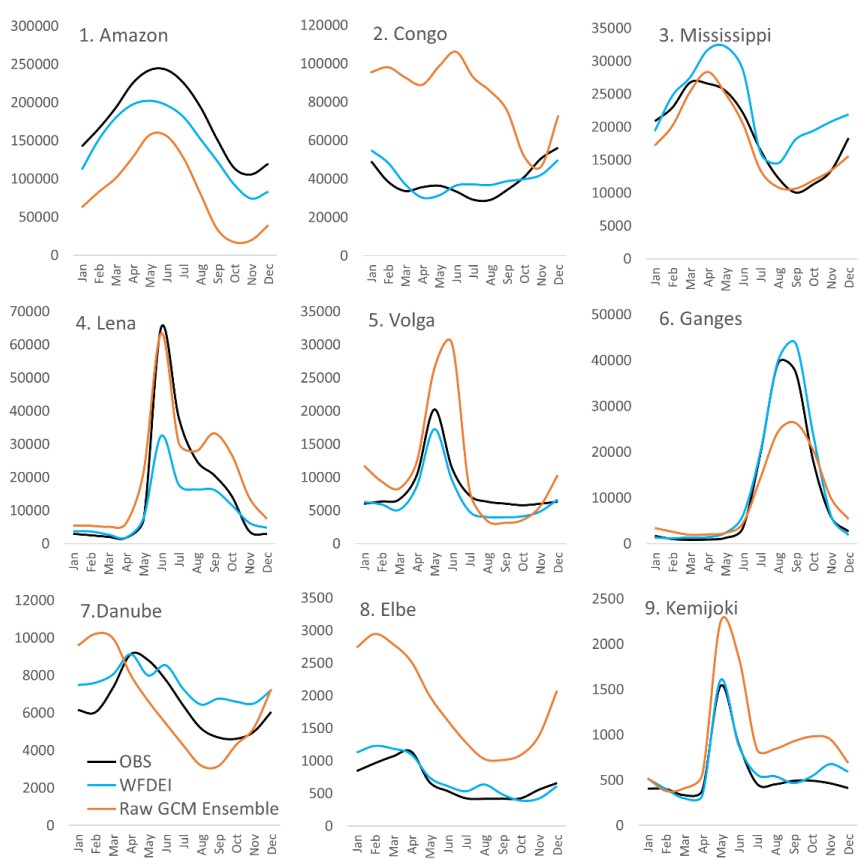

**Figure 5. Discharge seasonality [m³/s] derived from the period 1981-2010 for 9 study basins.**



| Indices | NSE | | PBIAS | | R2 | |
|---|---|---|---|---|---|---|
| Basins | WFDEI | GCM Ens | WFDEI | GCM Ens | WFDEI | GCM Ens |
| Amazon | 0.48 | -2.66 | -18.68 | -51.84 | 0.96 | 0.94 |
| Congo | 0.39 | -36.40 | 4.06 | 116.77 | 0.45 | 0.20 |
| Mississippi | 0.24 | 0.90 | 21.56 | -4.46 | 0.73 | 0.92 |
| Lena | 0.56 | 0.82 | -39.32 | 32.14 | 0.98 | 0.89 |
| Volga | 0.82 | -1.42 | -17.09 | 35.12 | 0.95 | 0.66 |
| Ganges | 0.94 | 0.80 | 19.48 | -9.51 | 0.99 | 0.91 |
| Danube | 0.28 | -1.51 | 15.20 | 1.14 | 0.88 | 0.19 |
| Elbe | 0.67 | -26.04 | 8.28 | 179.83 | 0.81 | 0.86 |
| Kemijoki | 0.91 | -0.98 | 8.55 | 66.50 | 0.94 | 0.89 |

**Figure 6. Comparison of evaluation metrics derived from seasonal runoff data. The simulations forced with WFDEI data and the raw GCM Ensemble data are compared with observed discharge data.**





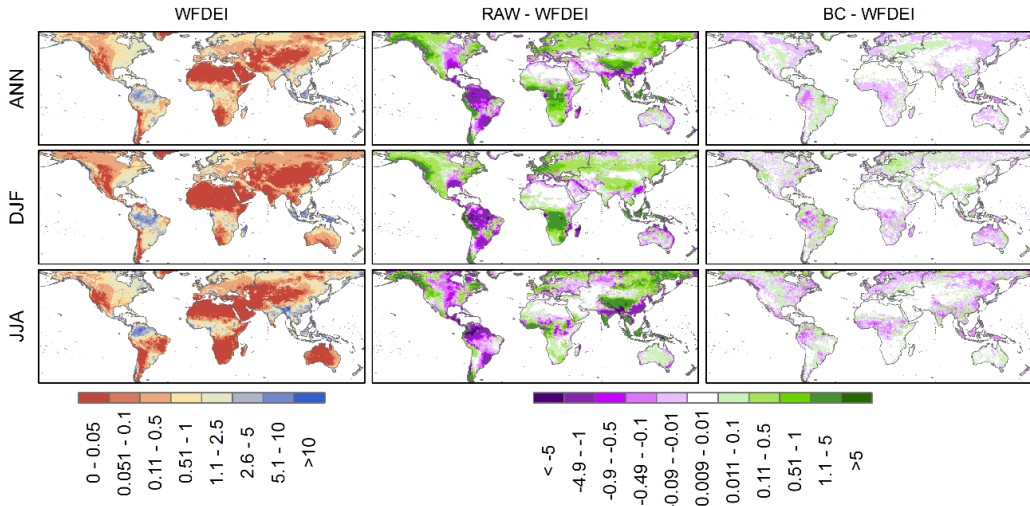

**Figure 7. Runoff production [mm/day], for WFDEI input forcing (left column), and differences in runoff forced the GCM ensemble and WFDEI, for raw (middle column) and bias adjusted (left column) forcing. Results are shown for ANN, DJF and JJA averages of the 1981-2010 period.**





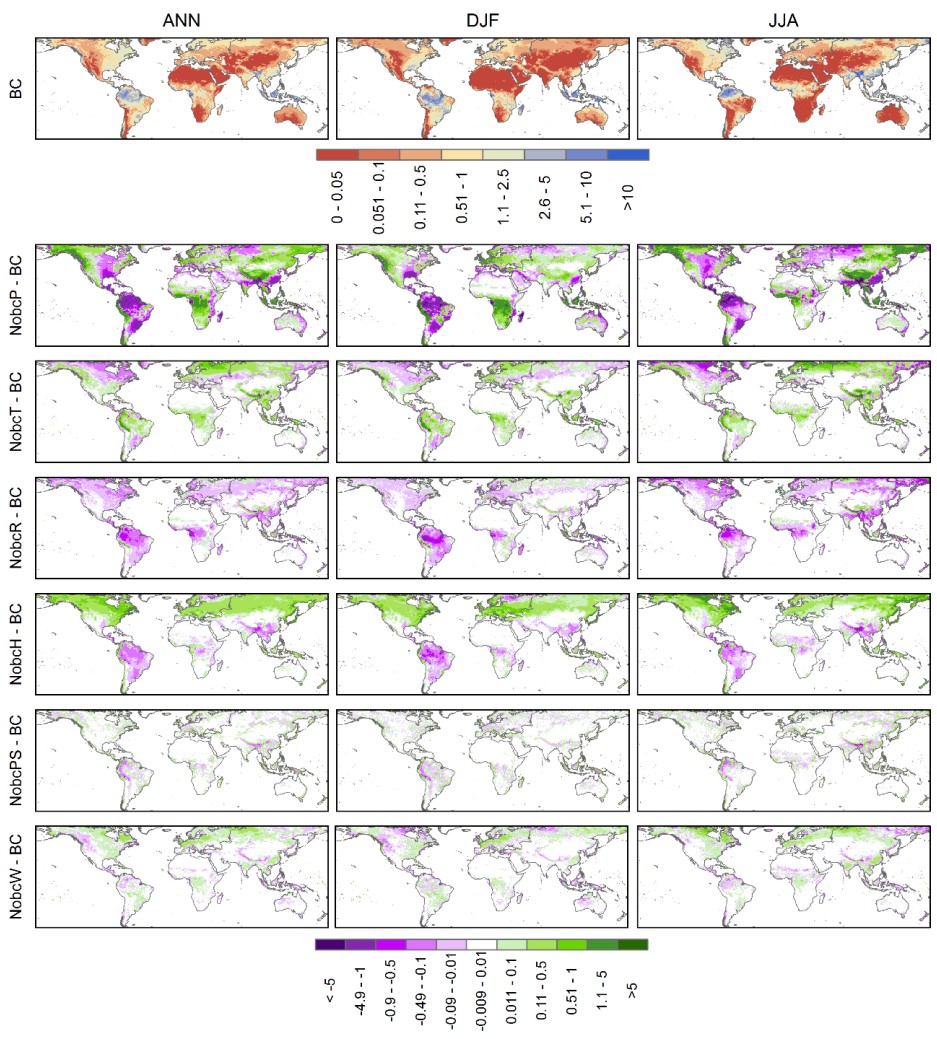

.

**Figure 8. Runoff production [mm/day], from all bias corrected GCM ensemble forcing (top row), and differences in runoff forced with one variable uncorrected and all bias corrected ensemble. Results are shown for ANN, DJF and JJA averages of the 1981-2010 period.**





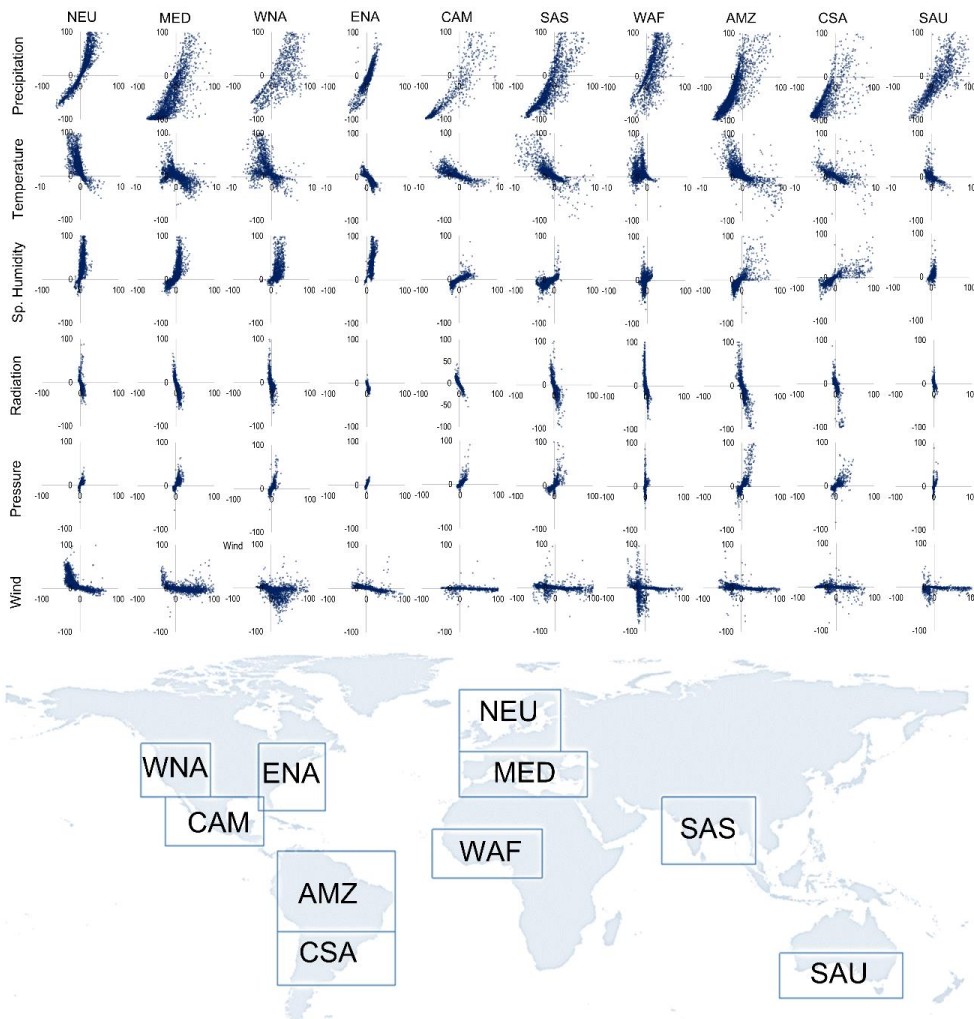

**Figure 9. Scatterplots of percent bias of a non-bias corrected variable and its effect on output runoff. Horizontal axis expresses the percent bias of the input variable and vertical axis expresses the change in output runoff caused by the input variable bias. Scatterplots are shown for the input variables P, Tas, Huss, Rlds/Rsds, Ps and Huss and for 10 selected land regions (top). Location and extent of the selected regions (bottom).**




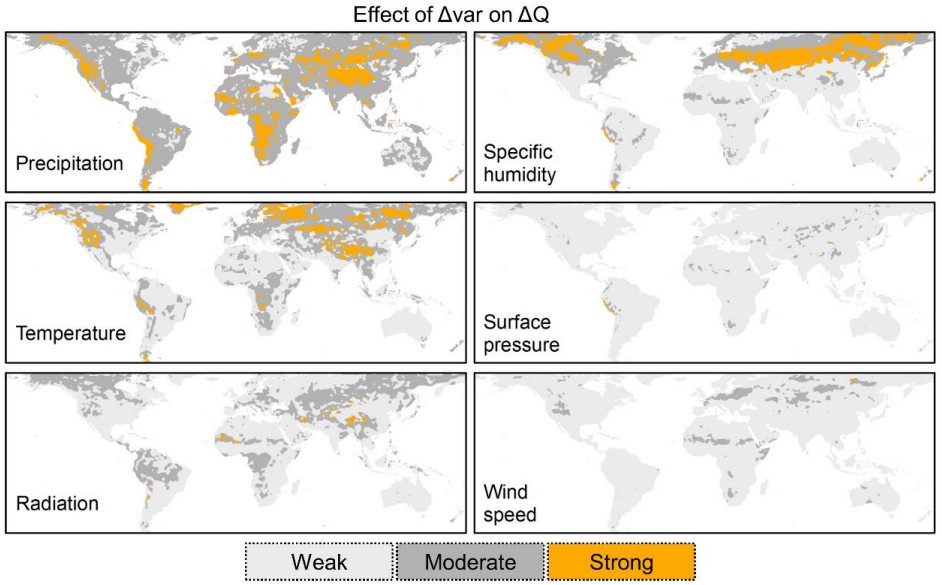

**Figure 10. Effect of the bias of each of the six forcing variables on output runoff.**





**Figure 11. As Figure 10, and additionally the areas where low (Q5) and high (Q95) runoff are strongly affected by the variables' biases.**