# Peer review of "Hotspots of sensitivity to GCM biases in global modelling of mean and extreme runoff."

_Hydrology and Earth System Sciences, 2016_

## Referee Comment (RC1) · Anonymous Referee #1 · 22 Nov 2016

The manuscript "Hotspots of sensitivity to GCM biases in global modeling of mean and extreme runoff", submitted by Papadimitriou and co-authors to the journal Hydrology and Earth System Sciences, investigates in general impacts of GCM biases to impact models, and specific to runoff. While in general the topic is of high relevance to the impact modelers, I see some major and (too) many minor drawbacks in the manuscript and I suggest a rejection. I have the impression that the present manuscript version was written quickly and not well reviewed by the co-authors. I see many analyses, which are in a way interesting, but I do not have the feeling that they are well discussed (and nearly nowhere interpreted) in order to provide benefit to the scientific audience. I had severe problems to get conclusions out of the paper for me, and also problems reviewing the content of the paper due to those numerous problems. The lack of consistency in wording, abbreviations and formatting (e.g. tables) does not fit at all to the

quality approach of HESS, even not for a submission to HESSD. As the general topic and some analyses are interesting, I would like to encourage the authors to shape the manuscript and to submit it again after careful revision. For this reason, I spend much time to list all my concerns in order to give some ideas for modification of the manuscript.

Major

1. I read the manuscript twice and intensively and then again the research questions. To me, the research questions are not sufficient formulated and answered in the results / discussions. I miss a clear rationale/storyline of this study. Is it to show hotspots of biases? Is it to show the effect of bias correction? Is it to show the effect of single bias corrected variables to runoff?

2. Wording / Definitions. I had many problems to understand the content due to vague and not consistent wording. I count at least 7 different names for naming variables like precipitation or temperature: "forcing parameters", "parameters", "meteorological variables", "climate data", "forcing variables", "fields", "forcing data". Is that all meant synonymously? Such vague wording makes it hard to understand the manuscript. I strongly suggest to strictly define names and use them consistently throughout the manuscript (this is a drastic example, but holds true for other definitions).

3. Text / Figures / Tables: As another example for previous comment – the specific variables are named not consistently. I suggest to define an common abbreviation at P4,L6, e.g. temperature (tas) and use the abbreviation throughout the document, also in figure / table headings etc.

4. Some basic and very important information are missing in the document. For example, I miss the information about the temporal (and spatial) resolution of the JULES model and specifically for the analyses. Such information is important for later interpretation e.g. at P6, L16.

[Figure]

5. It is a good idea to have an ensemble of model runs with different GCMs. Please provide why you use an ensemble, e.g. in the introduction. Furthermore, I suggest to show at least for one or two analyses also the spread among the runoff output among the GCMs. This would help the reader to interpret the effect of GCM choice and bias correction.

6. Results/discussion section. There is a large numbers of tables and figures (including the supplement) and probably out of that, the result of each analysis is presented very shortly. For tables, often by reproducing the numbers from the tables. A discussion / interpretation of the results is often too short (or missing completely).

Minor

- P 1, L13: "the physical consistency" – If I get it right you motivate your study at least in the abstract with the lack of information on bias correcting specific variables in terms of physical consistency. I do not see, how the present manuscript solves this problem. The partial correction assessment will lead to physical inconsistency, e.g. as variables are linked to each other (e.g. radiation and temperature).

- Abstract: given that your assessment is done with one specific impact model, you cannot write the conclusions in the abstract as this is a general rule. Impact models with different structures might react differently. Therefore I suggest to name the impact model in the abstract and relate the findings specificly to the model.

- Intro: Given the number of analyses in the manuscript, the introduction should be extended to cover more recent studies (if available) in general and more specific to guide the reader to the research questions. For example, an introduction into special problems of bias correction in terms of "extremes" is missing, what is meant with "past hydrologic indicators", why it is overall suitable to leave out "parameters" from a bias correction. A well formulated and structured information helps to guide the reader through the document. Furthermore, I suggest to add a short explanation of each research question.

- P2, L7: to cite a paper from 2007 regarding computing power limitations in a paper that is submitted in 2016 is a bit antiquated. Please check if this is still the case and provide a more recent reference (if so).

- P3,L8: You mention a focus on extreme events – isn't it a bit drastic formulated, if statistical high / low flows are assessed? Furthermore, those low/high flows are only one example in this manuscript, as mostly annual / seasonal values are the focus.

- P4,L30: Why did you use linear functions instead of gamma? Please provide some details of the bias correction method for non-BC experts. Impact modelers are interested e.g. if BC preserves trends, seasonality etc. Furthermore, I miss a statement if it is valid to use MSBC for other variables than precipitation.

- P6,L14: Please note, that common indicator for statistical low flow is Q95 and for high flows it is Q5. In hydrology, it is (mostly) defined as probability of exceedance, and Q95 means that the discharge Q is exceeded in 95% of the time, and acts as therefore as indicator for low flows. I suggest to use this common definition for a manuscript revision.

- P6,L22: Is each time step considered in the analysis (and how large is the time step)? Is there any time lag considered (e.g. precip to runoff)?

- P6,L24: I do not agree with similar usage of RF and Q. Q is defined as discharge (streamflow) whereas R (or as you define it as RF) is defined as runoff. Please do not use both terms synonymously. It is, e.g. at P6,L32 for me hard to judge if the effect is shown for runoff or discharge (accumulated runoff according to the drainage network), and that is to my impression important for interpreting the scatter plots.

- P7, Section 2.6: What exactly does "data of seasonal monthly (annual cycle)"? Is it the time series of monthly discharges from 1981 to 2010 (360 values)? Mean monthly (12) values? If the latter, does it make sense to use efficiency criteria then? How do you deal with gaps in GRDC data base? Where are the discharge observation stations

located?

- P8, Section 3: Why is it 3 and not 2.7? It belongs to methods section

- P8, Section 4.1: Suggest to clarify section heading to include annual or long term annual before bias.

- P9,L19: Could the pattern in surface pressure be related to the interpolation scheme from GCM resolution to those of JULES?

- P9,L31ff: reads partly like a repetition of the figure, please avoid this. You could, for example, give a statement how this pattern is among the 3 JULES runs driven by the single GCMs.

- P10,L30: It is interesting that Mississippi and Lena is best captured by raw GCM simulations. As this is a results/discussion section, I would encourage you to interpret / analyze, why this is the case. Is it due to JULES model, due to bias correction, due to GCM ensemble (how do single GCMs perform?) or due to basin-specific characteristics?

- P11,L5ff: this is something like an initial discussion. I suggest to use this to really discuss / interpret the results.

- P11,L12: I am not sure if it is correct to state the "remaining biases in output runoff". It is rather a difference in runoff due to bias correction (or no bias correction). The bias in runoff itself is not known.

- P11,L25ff: Rather than describing what a reader can see itself in the figure, you could assess e.g. global mean difference (or something like at xx % of land area, biases are > 1 mm/day) to give the reader more information.

- P12,L7: if P is dominant, what about humidity in northern regions? Isn't it more dominant (or is it just a question of interpreting colors?)

- P12,L11: I feel it is a bit too simplistic to relate biases in temperature to runoff overestimation. Isn't it an interrelation of variables? This part is again a simple repetition of the figure without additional information.

- P13,L12f: I think the meaning of "change" is different to "bias", or? Please use consistent wording.

- P13,L15: for what does dq stands for? Is it runoff, discharge? Not yet defined.

- P13,L18: I think it is rather simple to explain why there are so large differences in temperature impact to runoff in Northern Europe and Central America if you take over a geographic / climatic perspective. Please do so, it is a results and discussion part.

- P13,L31: please explain what you mean with "minimum median"

- P14,Sect. 4.7.: often a simple repetition of the numbers of Table 4, not useful.

- P15, Section 4.8: In general, such a study is of value, once it is done in a consistent way. One next step could be a multi-model assessment. Please include in caveats that you use an ensemble only. I think due to this, a large part of variation got "lost".

- P15,L27f: what do you mean with "past hydrologic indicators"?; suggestion to replace "hydrological applications and climate impact assessments" simply by "impact models"

- P16,L4: In the research aims you wanted to find out which parameters can be neglected. It is different to answer then the priorities of variables that needs to be bias corrected. And isn't it model dependent, which variables are needed at all, and which are most sensitive? I think such a generalization is not possible with the present study.

- P16, acknowledgments: why is the first part a quote?

- Reference list: please go through ref p16,l27 (upper/lower first names), p17,l30 (doi missing), p18,l6 (upper/lower first names), p18,l22,24 (page range). I did not checked if reference list is complete.

- Table 1: Value of table is questionable when a map is missing. Please revise table

caption to be more informative. General styling rule for tables: avoid vertical lines.

- Table 2: what does "Atm." stands for? Suggest to write °lon x °lat By the way, how did you dealt with the scale mismatch during bias correction?

- Table 3: In table caption you refer to "percent bias", but the variables are indicated by units like kg/kg. Not clear what is shown in the table. Some numbers are quite interesting, e.g. decrease in Tas and strong increase of runoff for WNA, NEU and others. What is the mechanism behind? Could be discussed.

- Table 4: meaning of the "+" not explained in caption. Still not sure if you mean low flow (= streamflow) or runoff.

- Figure 1: Please indicate abbreviations (P, T, etc) in caption. Use abbreviations consistently (RAW, BC). Note that in general figures and tables need to have a self-explaining caption in order to allow the reader to go through the figures and get the main message of the paper.

- Figure 2: why is the +60 at y-axis more distant from 50 than -60 from -50? Readability would be increased if dQ is written vertical on Y-axis (I had to think about some minutes what is shown on which axis), please revise caption to make it easy for the reader to follow. Interesting approach in general, but I guess sensitive on threshold. How sensitive are the results to thresholds of "strong"?

- Figure 3: Why is a dry bias in the Amazon colored in blue and not in red (which would be more intuitive)? Please define exactly what you mean with radiation components (downward? upward?). Figure caption needs to be revised. I do not see annual averages (i.e. absolute values), I see difference maps.

- Figure 4: Would be interesting to see the same figure also for BC values, I still do not have a good impression how the bias correction works for seasonal values. Add a column for global land area also. As a map for all Giorgi regions are missing, hard to interpret for a reader. Please modify caption, e.g. to proper cite the 24 regions.

- Figure 5: If I get it right, mean monthly values are shown for the 9 basins. Please define the gaging stations of the basins. In general, provide details about the basins (not sure if everybody is familiar with Kemijoki basin). Furthermore, revise line graph. You have 12 data points but due to the fitting function you use, the reader get the impression that you show e.g. daily values. Use linear interpolation instead. Y-Axis label is missing. Define legend more careful or use caption to explain it. A reader could think that WFDEI forcing provides runoff which is displayed here, but in fact it is JULES driven by WFDEI.

- Figure 6: what is meant with "seasonal runoff data"? How many data points were used to calculate metrics? Hard to believe that GCM ens gets NSE of 0.8 and a relatively low PBIAS when looking at Fig 5. What does the color hue and saturation mean?

- Figure 7/8: It is very interesting to see that bias correction might slightly change the sign of difference compared to WFDEI. I suggest to discuss that in more detail in the manuscript. For my feeling, too many colors in the difference maps, very hard to distinguish it also due to the given extreme small map size. Revise figure caption: bias adjusted is shown at right column. Combine Fig. 7 and 8. What is meant with "runoff production"?

- Figure 8: Indicate abbreviations: NobcR? NobcH? Etc.

- Figure 9 / Supplement S5: To be honest, I do not have an idea what is represented with each dot. Is it a grid cell and the mean of the time series, or is it each time step for the spatial average? Maybe this is a stupid question, but I could not find it out. In addition - is it really necessary to repeat the entries from Fig 9 in the figure S5 of the supplement? Due to the horizontal labels of variables, I first got the impression that e.g. Precipitation is shown at Y-Axis and Runoff at X-Axis but that does not fit to caption. Please revise. Is everywhere a % difference shown? Why is scale for Temperature +-10 %, for the others +-100%? I suggest to show at Fig 9 (or a separate one) all 24 Giorgi & Bi 2005 regions. Provide reference to data source of the region definition.

Revise figure caption. You mention abbreviations but do not use them in the figure.

- Figure 10. Figure caption. What is "output runoff"? do you mean mean runoff of JULES driven with ...? Please provide a meaningful figure caption.

- Figure 11. Interesting approach to add the regions. It seems that there is no difference between Weak and Moderate? Combine Fig 10 and 11 to save space.

- Figures Supplement: Revise captions (formatting, consistency), e.g. bias adusted in right column at S3 (but also others).

- Figure S3,S4,S6, Table S1: not referenced in main text, why shown in Supplement?

Technical issues:

- Please take care that every abbreviation is defined at its first occurrence (e.g. GCM p1, l20).

- P2,L17: take care of correct wording: "climate model output data" instead of "climate model data", or?

- P2,L19: sentence with "Typically..." Are there other bias correction methods available (that do not use observations – that must be available only for historical period)? This sentence is unclear to me

- P2,L27: What do you mean with "biophysical impact model"?

- P2,L28: Check correct name and abbreviation of ISIMIP (www.isimip.org)

- P2,L29: Not all of the participating models need e.g. humidity, surface pressure and wind speed, that depends e.g. on the equation of potential evapotranspiration.

- P4,L6: please define if longwave/shortwave downward or upward or net radiation is meant.

- P4,L16: Does the citation truly describes the Penman-Monteith-Method?

- P4, L23: What is MSBC?

- P4, L26: what is CDF?

- P5,L28f: does "uncorrected" refer to RAW and "corrected" to BC? If so, please use abbreviations that you have defined earlier.

- P8,L31: There is no "degrees K"

- P8,L22: please indicate if "annual" or "mean" biases are meant.

- P12,L22: should be runoff underestimation, or?

- P14,L8: r in fraction is missing

---

## Referee Comment (RC2) · Anonymous Referee #2 · 2 Dec 2016

General comments:

This paper examines the impact of GCM biases on estimates of mean runoff as well as extreme low and high flows. The authors examine a small ensemble of GCMs (3 models) and use one land-surface model, the Joint UK Land Environment Simulator (JULES) model. It is well known that biases in GCM output, particularly precipitation and temperature impact hydrological model simulations as the authors state. The premise of this paper is to examine the impact bias correction has on modeled runoff for the forcing variables needed to run a model in energy balance model. They highlight that precipitation has the largest impact on runoff, but also that other variables impact runoff in a potentially significant manner.

I find the paper needs substantial English editing before it would be acceptable for

publication. Additionally, the text of the paper offers little thoughtful discussion of the results, it generally just restates values from tables or general patterns in the Figures. I also find some of the results to be highly suspect, the authors need to recheck their model simulations very closely. Overall, the idea of a systematic examination of bias correction on model runoff is interesting, but the execution is lacking. Therefore I recommend rejection.

Specific comments:

1) Throughout the article: 1) abbreviations of variable names; 2) acronyms; and 3) units, are 1) hard to follow or inconsistently used (e.g. runoff is RF instead of R, and then grid cell runoff is interchanged with Q, which is typically used for stream discharge); 2) not defined before use such as MSBC; or 3) not used properly (e.g. Table 3). This goes along with the need for substantial English editing for clarity, grammar, etc.

2) The design of the experiments in the partial bias correction assessment could be done differently to increase their utility. For example, it is unrealistic to think users are going to bias correct every variable except temperature. I think an iterative addition of variables from no bias correction would show more useful results. Stepping through bias correction for only P, only P and T, only P, T and humidity, etc. may be more useful. This would show the added value of additional bias correction over the previous iteration.

3) Many of the sections just restate values from Tables and Figures without any discussion and/or insight given.

a) Section 4.1 is very long for the simple task of stating that the bias correction works, yet it just steps through the figures and does not discuss why the remaining biases are where they are.

b) Section 4.2 is interesting, the authors should see Gutmann et al. (2014) for further

discussion on changes in bias correction statistics across spatial scales.

c) There is no discussion of the differences between raw and bias corrected in section 4.3 or the regions where the raw fields are better than the observations forcing. This has implications for bias correction to observed products. We may be actually making things worse in regions with poor quality or uncertain observations.

4) The results in section 4.5 and 4.6 should be investigated in much more detail. Specifically, there seems to be a large response to bias corrected humidity in the boreal forest regions across all of North America (Figures 8-10). Why and how does this happen? I'm questioning that a change in humidity could increase runoff by up to several tenths of a mm/day, or potentially up to 100 mm/year. That is a significant fraction of the annual precipitation in many of these regions. The differences between raw and the observations (Fig. 3) show only modest corrections, roughly 0.01 to 0.1 g/kg, is that enough extra water, or what else changes in the system? Is the bias correction creating supersaturated conditions, which then condenses on the trees and creates canopy throughfall and increased surface water input? Does the model forest use less water if it is more humid?

From Figure 9 and Table 3 in the NEU region, a 4% change in humidity results in an astounding 25% increase in runoff, for an extreme sensitivity to humidity of 6.25x. Also, the sensitivities (elasticities) to temperature stated on page 13, line 18 are very large as well. Elasticity work across the US and China (e.g. Fu et al. 2007; Vano et al. 2012) show potentially less sensitivity to temperature depending on the region and model. Finally there has been much work on precipitation sensitivity that is not referenced at all and used as a comparison point (e.g. Sankarasubramanian et al. 2001).

This past work should be discussed in the context of this study and used as checks on these simulations for realism.

References: Fu, G., S. P. Charles, and F. H. S. Chiew, 2007: A two-parameter climate elasticity of streamflow index to assess climate change effects on annual streamflow.

Water Resour. Res., 43, W11419, doi:10.1029/2007WR005890.

Gutmann, E. D., and co-authors, 2014: An intercomparison of statistical downscaling methods used for water resource assessments in the United States. Water Resour. Res., 50, 7167-7186.

Sankarasubramanian, A., R. M. Vogel, and J. F. Limbrunner, 2001: Climate elasticity of streamflow in the United States. Water Resour. Res., 37, 1771–1781.

Vano, J. A., T. Das, and D. P. Lettenmaier, 2012: Hydrologic sensitivities of Colorado River runoff to changes in precipitation and temperature. J. Hydrometeorol., 13, 932-949.